# Inhibition of HDAC and Signal Transduction Pathways Induces Tight Junctions and Promotes Differentiation in p63-Positive Salivary Duct Adenocarcinoma

**DOI:** 10.3390/cancers14112584

**Published:** 2022-05-24

**Authors:** Masaya Nakano, Kizuku Ohwada, Yuma Shindo, Takumi Konno, Takayuki Kohno, Shin Kikuchi, Mitsuhiro Tsujiwaki, Daichi Ishii, Soshi Nishida, Takuya Kakuki, Kazufumi Obata, Ryo Miyata, Makoto Kurose, Atsushi Kondoh, Kenichi Takano, Takashi Kojima

**Affiliations:** 1Department of Cell Science, Research Institute for Frontier Medicine, Sapporo Medical University School of Medicine, Sapporo 060-8556, Japan; masayanakano0606@gmail.com (M.N.); kizuku020@gmail.com (K.O.); shindo.y@sapmed.ac.jp (Y.S.); t.konno1225@gmail.com (T.K.); kohno@sapmed.ac.jp (T.K.); d10660348@yahoo.co.jp (D.I.); sossoshi9@gmail.com (S.N.); 2Department of Otolaryngology, Sapporo Medical University School of Medicine, Sapporo 060-8556, Japan; kurokuma1029@yahoo.co.jp (T.K.); obata4416@gmail.com (K.O.); ryomiyata1103@yahoo.co.jp (R.M.); mkurose0411@gmail.com (M.K.); akondosa@sapmed.ac.jp (A.K.); kent@sapmed.ac.jp (K.T.); 3Department of Thoracic Surgery, Sapporo Medical University School of Medicine, Sapporo 060-8556, Japan; 4Department of Anatomy, Sapporo Medical University School of Medicine, Sapporo 060-8556, Japan; ksin@sapmed.ac.jp; 5Department of Pathology, Sapporo Medical University School of Medicine, Sapporo 060-8556, Japan; auditt3200@gmail.com

**Keywords:** salivary duct adenocarcinoma, tight junctions, p63, HDAC inhibitors, EW-7198, SP600125

## Abstract

**Simple Summary:**

The p53 family p63 gene is essential for the proliferation and differentiation of various epithelial cells, and it is overexpressed in some salivary gland neoplasia. Histone deacetylases (HDACs) are thought to play a crucial role in carcinogenesis, and HDAC inhibitors downregulate p63 expression in cancers. p63 is not only a diagnostic marker of salivary gland neoplasia, but it also promotes the malignancy. Inhibition of HDAC and signal transduction pathways inhibited cell proliferation and migration, induced tight junctions, and promoted differentiation in p63-positive salivary duct adenocarcinoma (SDC). It is, therefore, useful in therapy for p63-positive SDC cells.

**Abstract:**

Background: The p53 family p63 is essential for the proliferation and differentiation of various epithelial basal cells. It is overexpressed in several cancers, including salivary gland neoplasia. Histone deacetylases (HDACs) are thought to play a crucial role in carcinogenesis, and HDAC inhibitors downregulate p63 expression in cancers. Methods: In the present study, to investigate the roles and regulation of p63 in salivary duct adenocarcinoma (SDC), human SDC cell line A253 was transfected with siRNA-p63 or treated with the HDAC inhibitors trichostatin A (TSA) and quisinostat (JNJ-26481585). Results: In a DNA array, the knockdown of p63 markedly induced mRNAs of the tight junction (TJ) proteins cingulin (CGN) and zonula occuludin-3 (ZO-3). The knockdown of p63 resulted in the recruitment of the TJ proteins, the angulin-1/lipolysis-stimulated lipoprotein receptor (LSR), occludin (OCLN), CGN, and ZO-3 at the membranes, preventing cell proliferation, and leading to increased cell metabolism. Treatment with HDAC inhibitors downregulated the expression of p63, induced TJ structures, recruited the TJ proteins, increased the epithelial barrier function, and prevented cell proliferation and migration. Conclusions: p63 is not only a diagnostic marker of salivary gland neoplasia, but it also promotes the malignancy. Inhibition of HDAC and signal transduction pathways is, therefore, useful in therapy for p63-positive SDC cells.

## 1. Introduction

Salivary gland cancer is a rare malignancy that includes more than 20 subtypes according to the WHO 2017 classification of head and neck cancers [1,2]. While head and neck cancers are predominantly squamous cell carcinomas, salivary gland cancers are predominantly adenocarcinomas. The standard treatment is surgery, with additional postoperative adjuvant therapy in cases of locally advanced cancers. These treatments allow for good local control rates, but a large number of patients develop distant metastasis and, as a result, the overall and disease-free survival rates decrease [3]. Recently, analysis of therapeutic target molecules for salivary gland duct adenocarcinoma such as HER2 has been reported. HER2 is specifically expressed in salivary gland duct adenocarcinoma, which is the most malignant form of salivary gland cancer [4]. However, since standard non-surgical treatments are ineffective, further studies of new drugs are crucial.

The p53 family p63 gene is essential for the proliferation and differentiation of epithelial cells [5]. The p63 gene has two distinct isoforms, proapoptotic TAp63 and oncogenic ΔNp63 [6]. It is reported that the frequency of p63 positivity is high in squamous cell carcinoma, including in human head and neck squamous cell carcinoma (HNSCC), thymic tumors, urothelial carcinoma, and various salivary gland tumors [7]. ΔNp63 is known as an oncogenic driver in squamous cell carcinoma and plays an important role in HNSCC cell survival, suppressing p73-dependent apoptosis [8,9]. ΔNp63 can also regulate the expression of genes correlated with epithelial-to-mesenchymal transition (EMT) [10]. Epithelial-to-mesenchymal transition is the mechanism via which cancer cells lose polarity, separate from each other and acquire the characteristics of a mesenchymal phenotype. Cancer cells with mesenchymal traits increase cell migration and invasion, resulting in cancer metastasis [11]. The human TP63 gene is aligned with histone H3 acetylated at lysine 27 (H3K27Ac) [12]. p63/p40 (ΔNp63) is used for diagnostic utility in the histologic differentiation of salivary gland tumors [13]. However, the roles and regulation of p63 are unknown in salivary gland tumors.

Transforming growth factor β (TGF-β) superfamily signaling has been implicated in driving developmental programs, cellular proliferation, and differentiation [14,15]. TGF- β1 promotes the migration and invasion of salivary adenoid cystic carcinoma [16]. Tumor necrosis factor (TNF) receptor-associated factor 6 (TRAF6) regulates TGF-β-mediated salivary adenoid cystic carcinoma (SACC) progression through SMAD2/3-ERK-p38-JNK cascades [17]. p63 maintains salivary gland stem and progenitor cell proliferation via TGF-β signaling [18]. EW-7197 (Vactosertib) is an orally available inhibitor of the kinase activity of TGF-β type I receptor/ALK-5 [19]. The safety and efficacy of EW-7197 against different cancer types have been reported in a number of trials [20]. However, the effects of EW-7197 for salivary gland tumors are unknown.

The histone deacetylase (HDAC) enzymes, which regulate the reversible acetylation of lysine residues of histones and non-histone proteins, are epigenetic regulators of the transcriptional activities of certain genes [21,22]. In addition, HDACs regulate cell cycle progression, cell survival, and differentiation and disruption of HDAC activity, which may be associated with malignancy in various cancers [23]. The following four classes of the HDAC family are currently known: class I HDACs (HDAC 1-3, and 8), class II HDACs (HDAC 4–7, 9, and 10), class IV (HDAC 11), and class III (sirtuin family: SIRT1-7) [21,23]. To date, different HDACs have been shown to be abundantly expressed in various human cancers such as laryngeal, gastric, liver, colon, breast, lung, and salivary gland cancers [24,25,26]. In salivary gland tumors, the expression of HDAC6 is correlated with a poor prognosis [26].

HDAC inhibitors are epigenetic regulators that relax chromatin structures through increased acetylation of histones and, as a result, induce cell cycle arrest, differentiation, and the apoptosis of malignant cells [27,28,29]. Furthermore, HDAC inhibitors, when combined with known therapies, inhibit resistance to antitumor drugs [30]. Therefore, HDAC inhibitors demonstrate the potential to be a new major treatment that exhibits antitumor activities. Trichostatin A (TSA), which is a strong and specific inhibitor of class I and II HDACs, exerts antitumor effects in various cancers, such as those of the breast, bladder, and head and neck [31,32,33]. TSA also induces reverse EMT and, as a result, decreases cell migration and invasion in colorectal cancer [34]. Quisinostat (JNJ-26481585) is a novel second-generation HDAC inhibitor with the highest potency against HDAC1 and an orally bioavailable anticancer drug [35]. It is reported that quisinostat suppresses cell proliferation in hepatocellular carcinoma, induces cell cycle arrest with upregulation of p53, and decreases cell migration by inhibiting EMT in lung cancer [36,37]. Thus, HDAC inhibitors are anticancer agents that have attracted a great deal of attention recently.

Tight junctions (TJs) are adhesion structures that exist on the most apical side of the intercellular space between epithelial cells [38,39]. It is also well-known that differentiated epithelial cells are characterized by apical TJs [40]. TJs are composed of membrane proteins such as claudin (CLDN), occludin, and junctional adhesion molecules (JAMs), and scaffold proteins such as zonula occludens. The scaffold proteins bind to regulator proteins, such as cingulin (CGN), and cytoskeletal proteins, and, as a result, connect the scaffold with microtubules and actin [41,42,43]. In addition to these TJ proteins, tricellular tight junction (tTJ) proteins such as the angulin family, including angulin-1/lipolysis-stimulated lipoprotein receptor (LSR) and tricellulin, play a role in the barrier function of epithelial cells [44]. Furthermore, TJ proteins play an important role in not only epithelial barrier function but also in the regulation of gene expression and signal pathways [45,46].

c-Jun N-terminal kinase (JNK), known as a stress-activated protein kinase, regulates normal epithelial biological processes, including the assembly of adherens and tight junctions, and is involved in the regulation of tTJs [47]. The JNK inhibitor SP600125 affects epithelial barrier and cell migration function via TJs in both normal and cancer cells [48,49].

CGN interacts with TJ proteins ZO-1, ZO-2, ZO-3, and AF6 [50]. CGN is also a TJ protein that binds to both actin filaments and microtubules (MTs), and localizes to centrosomes [51,52]. Furthermore, the planar apical network of non-centrosomal MTs is associated with TJs, including CGN, and the CGN may organize the formation of the apical MT network [51]. The HDAC inhibitor sodium butyrate upregulates the levels of protein and mRNA for CGN in Rat-1 fibroblasts, COS-7 cells, and HeLa cells [53].

In the present study, to investigate the roles and regulation of p63 in SDC, human SDC A253 cells were transfected with siRNA-p63 or treated with the HDAC inhibitors trichostatin A (TSA) and quisinostat (JNJ-26481585) under pretreatment with TGF-β type I receptor inhibitor EW-7197 and JNK inhibitor SP600125. They were then compared to primary cultured human salivary gland duct epithelial (HSDE) cells.

## 2. Materials and Methods

### 2.1. Reagents and Antibodies

Trichostatin A (TSA) was obtained from Sigma-Aldrich (St. Louis, MO, USA). Quisinostat (JNJ-26481585) was obtained from Selleck Chemicals (Houston, TX, USA). A TGF-β receptor type 1 inhibitor (EW-7197) was obtained from Cayman Chemical (Ann Arbor, MI, USA). A JNK inhibitor (SP600125) was obtained from the Calbiochem-Novabiochem Corporation (San Diego, CA, USA). A mouse monoclonal anti-p63 (clone DAK-p63) antibody was obtained from DAKO (Tokyo, Japan). A rabbit polyclonal anti-p63 antibody was obtained from Abcam (Cambridge, MA, USA). A rabbit polyclonal anti-LSR antibody was obtained from Novus Biologicals (Littleton, CO, USA). Rabbit polyclonal anti-occludin (OCLN), anti-cingulin (CGN), anti-zonula occludin-3 (ZO-3), anti-tricellulin (TRIC), anti-claudin (CLDN)-1, -4, and -7 antibodies were obtained from Zymed Laboratories (San Francisco, CA, USA). Mouse monoclonal anti-acetylated tubulin, anti-cytokeratin 5, and a rabbit polyclonal anti-actin antibody were obtained from Sigma-Aldrich (St. Louis, MO, USA). A mouse monoclonal antitubulin antibody was obtained from Santa Cruz Biotechnology (Dallas, TX, USA). Rabbit polyclonal anti-olfactomedin-like 3 (OLFML3) and anti-cytochrome P450 1A1/2 (CYP1A1) antibodies were obtained from Abcam (Cambridge, UK). rabbit polyclonal anti-phosphorylated MAPK and rabbit polyclonal anti-phosphorylated NF-κB antibodies were obtained from Cell Signaling Technology (Danvers, MA, USA). A monoclonal mouse anti-α-tubulin antibody was obtained from FUJIFILM Wako Chemicals Corporation Japan (Osaka, Japan). Alexa 488 (green)-conjugated anti-rabbit IgG and Alexa 594 (red)-conjugated anti-mouse IgG antibodies were obtained from Molecular Probes, Inc. (Minneapolis, MN, USA). The ECL Western blotting system was obtained from GE Healthcare UK, Ltd. (Buckinghamshire, UK).

### 2.2. Cultures of Cell Lines and Treatments

A253 cells derived from epidermoid carcinoma of the submaxillary salivary gland were purchased from ATCC (Manassas, VA, USA) and cultured in minimum essential medium (Sigma-Aldrich), supplemented with 10% fetal bovine serum (FBS, Invitrogen; Carlsbad, CA, USA), 100 U/mL penicillin, 100 μg/mL streptomycin, and 50 μg/mL amphotericin B. The cells were plated on 35 mm and 60 mm culture dishes coated with rat tail collagen (500 μg of dried tendon/mL in 0.1% acetic acid), and incubated in a humidified 5% CO_2_ incubator at 37 °C. The cells were transfected with the siRNA of p63 for 24 h and then treated with a TGF-β receptor type 1 inhibitor (EW-7197) and JNK inhibitor (SP600125) at 10 μM for 24 h. Some cells were treated with HDAC inhibitors TSA or JNJ at 1 μM and 10 μM, respectively, with EW-7198 and SP600125 at 10 μM for 24 h.

### 2.3. Isolation and Culture of Human Salivary Gland Duct Epithelial Cells

Cultured human salivary gland duct epithelial (HSDE) cells were derived from salivary gland tissues of patients with IgG4-related disease who underwent a sialoadenectomy at Sapporo Medical University. This study was approved by the ethics committee of the institution.

Human salivary gland tissues were minced into pieces (2–3 mm^3^) and washed with PBS containing 100 U/mL penicillin and 100 μg/mL streptomycin (Lonza Walkersville, Walkersville, MD, USA) three times. Minced tissues were suspended in 10 mL of Hanks’ balanced salt solution with 0.5 μg/mL DNase I and 0.04 mg/mL Liberase Blendzyme 3 (Roche, Basel, Switzerland). They were then incubated with a bubbling of mixed O_2_ gas containing 5.2% CO_2_ at 37 °C for 10 min. The dissociated tissues were subsequently filtered with 300 μm mesh, followed by filtration with 70 μM mesh (Cell Strainer; BD Biosciences, San Jose, CA, USA). After centrifugation at 1000× *g* for 4 min, isolated cells were cultured in bronchial epithelial basal medium (BEBM; Lonza Walkersville) containing 10% fetal bovine serum (FBS; CCB, Nichirei Bioscience, Tokyo, Japan) and supplemented with BEGM SingleQuots (Lonza Walkersville; including 0.4% bovine pituitary extract, 0.1% insulin, 0.1% hydrocortisone, 0.1% gentamicin, amphotericin-B (GA-1000), 0.1% retinoic acid, 0.1% transferrin, 0.1% triiodothyronine, 0.1% epinephrine, and 0.1% human epidermal growth factor), 100 U/mL penicillin, and 100 μg/mL streptomycin on 60 mm culture dishes (Corning Life Sciences, Acton, MA, USA) coated with rat tail collagen (500 μg of dried tendon/μL of 0.1% acetic acid). Following the above protocol, tissue dissociation and cell isolation were repeated for the same sample a maximum of seven times. The cells were placed in a humidified 5% CO_2_:95% air incubator at 37 °C. The retroviral vector BABE-hygro-hTERT was used. The viral supernatant was produced from an ecotropic packaging cell line by transfection with plasmid DNA. The packaging cells were cultured in Dulbecco’s modified Eagle’s medium, containing 10% FBS and supplemented with 100 U/mL penicillin and 100 μg/mL streptomycin. At 24 h after plating on 60 mm dishes, HSDE cells in the primary culture were exposed overnight to the viral supernatant containing the retrovirus. After being washed with serum-free BEBM medium, the hTERT-transfected HSDE cells were cultured in serum-free BEBM medium supplemented with the above-mentioned factors and 2.5 μg/mL amphotericin B. The cells grew to confluence on the 60 mm culture dishes within 1–2 weeks, and the first passage was carried out using 0.05% trypsin-EDTA (Sigma-Aldrich, St. Louis, MO, USA) in 60 mm culture dishes. On day 5 after the first passage, the second passage was carried out in the same manner in 60 or 35 mm culture dishes, and cells at the second passages were used for experiments on days 5–7 after plating. HSDE cells were transfected with siRNAs of p63, CGN, and LSR for 48 h.

### 2.4. GeneChip Analysis

Microarray slides were scanned using a 3D-Gene human Oligo chip 25k. (TORAY, Tokyo, Japan), and the microarray images were automatically analyzed using AROSTM, version 4.0 (Operon Biotechnologies, Tokyo, Japan).

### 2.5. RNA Interference and Transfection

siRNA duplex oligonucleotides against p63, CGN, and LSR were synthesized by Thermo Fisher Scientific (Waltham, MA, USA). The sequences were as follows: siRNA of p63A (sense: 5′-GCACACAAUUGAAACGUACAGGCAA-3′; antisense: 5′-UUGCCUGUACGUUUCAAUUGUGUGC-3′), siRNA of p63-B (sense: 5′-ACCAU-GAGCUGAGCCGUGAAUUCAA-3′; antisense: 5′-UUGAAUUCAC-GGCUCAGCUCAUGGU-3′), siRNA of LSR (sense: 5′-CCCACGCAACCCAUCGUCAU-CUGGA-3′; antisense: 5′-UCCAGAUGACGAUGGGUUGCGUGGG-3′), and siRNA of CGN (sense: 5′-CCCACCAUGCUGCAGUUCAAAUCAA-3′; antisense: 5′-UUGAU-UUGAACUGCAGCAUGGUGGG-3′). At 24 h after plating, A253 cells and HSDE cells were transfected with siRNA of p63, LSR, or CGN using Lipofectamine™ RNAiMAX reagent (Invitrogen) for 48 h. A scrambled siRNA sequence (BLOCK-iT Alexa Fluor fluorescent, Invitrogen) was employed as the control siRNA.

### 2.6. Immunohistochemical Analysis

Human salivary gland tissues were obtained from patients with IgG4-related disease who underwent a sialoadenectomy at Sapporo Medical University. Informed consent was obtained from all patients, and this study was approved by the ethics committee of the institution.

The tissues were embedded in paraffin after fixation with 10% formalin in PBS. Briefly, 5 μm-thick sections were dewaxed in xylene, rehydrated in ethanol, and heated with Vision BioSystems Bond Max using ER2 solution (Leica BIOSYSTEMS, Tokyo, Japan) in an autoclave for antigen retrieval. Endogenous peroxidase was blocked by incubation with 3% hydrogen peroxide in methanol for 10 min. The tissue sections were then washed twice with Tris-buffered saline (TBS) and pre-blocked with Block Ace for 1 h. After washing with TBS, the sections were incubated with anti-p63 (1:200), anti-CGN (1:400), anti-ZO-3 (1:400), and anti-OLFML3 (1:400) antibodies for 1 h. The sections were then washed three times in TBS and incubated with Vision BioSystems Bond Polymer Refine Detection kit DS9800. After three washes in TBS, a diamino-benzidine tetrahydrochloride working solution was applied. Finally, the sections were counterstained with hematoxylin.

### 2.7. Transmission Electron Microscopy Analysis

A253 cells were cultured to confluence in 8 chambers of CultureSlides (FALCON). For transmission electron microscopy (TEM), the cultured cells were fixed in 2.5% glutaraldehyde in PBS overnight at 4 °C, followed by post fixing in 2% osmium tetroxide in the same buffer. Then, the cells were dehydrated with a graded ethanol series and embedded in Epon 812. Ultrathin sections were cut on a Sovall Ultramicrotome MT-5000. The sections were stained with uranyl acetate followed by lead citrate and examined at 80 kV with a transmission electron microscope (H7500; Hitachi, Tokyo, Japan).

### 2.8. Western Blot Analysis

Cultured cells were scraped from a 35 mm dish containing 400 μL of buffer (1 mM NaHCO_3_ and 2 mM phenylmethylsulfonyl fluoride), and HSDE cells were scraped from a 60 mm dish containing 300 μL of buffer, collected in microcentrifuge tubes, and then sonicated for 10 s. The protein concentrations of the samples were determined using a BCA protein assay reagent kit (Pierce Chemical Co.; Rockford, IL, USA). Aliquots of 15 μL of protein/lane for each sample were separated by electrophoresis in 5–20% SDS polyacrylamide gels (Wako, Osaka, Japan), before being electrophoretically transferred to a nitrocellulose membrane (Immobilon; Millipore Co.; Bedford, UK). The membrane was saturated for 30 min at room temperature with blocking buffer (25 mM Tris, pH 8.0, 125 mM NaCl, 0.1% Tween-20, and 4% skim milk), and incubated with anti-p63, anti-CGN, anti-ZO-3, anti-OLFML3, anti-CYPLA1, anti-LSR, anti-TRIC, anti-CLDN-1, -4, -7, anti-acetylated-tubulin, anti-pMAPK, and anti-actin antibodies (1:1000) at room temperature for 1 h or overnight. It was then was incubated with HRP-conjugated anti-mouse and anti-rabbit IgG antibodies at room temperature for 1 h. The immunoreactive bands were detected using an ECL Western blotting system.

### 2.9. Immunoprecipitation and Western Blot Analysis

A253 cells in 60 mm dishes were washed with PBS and 300 μL of NP-40 lysis buffer (50 mM Tris-HCl, 2% NP-40, 0.25 mM sodium deoxycholate, 150 mM NaCl, 2 mM EGTA, 0.1 mM Na_3_VO_4_, 10 mM NaF, and 2 mM PMSF) was added to the dishes. The cells were scraped off, collected in microcentrifuge tubes, and then sonicated for 10 s. Cell lysates were incubated with protein A-Sepharose CL-4B (Pharmacia LKB Biotechnology; Uppsala, Sweden) for 1 h at 4 °C, and then clarified by centrifugation at 15,000× *g* for 10 min. The supernatants were incubated with the polyclonal anti-CGN antibody bound to protein A Sepharose CL-4B overnight at 4 °C. After incubation, immunoprecipitates were washed extensively with the same lysis buffer and subjected to Western blot analysis with anti-CGN, anti-acetylated-tubulin, anti-LSR, anti TRIC, anti-CLDN-1, -4, -7, and anti-actin antibodies.

### 2.10. Immunocytochemistry

A253 cells and HSDE cells grown in 35 mm glass-coated wells (Iwaki, Chiba, Japan) were fixed with cold acetone and ethanol (1:1) at –20 °C for 10 min. After rinsing in PBS, the cells were incubated with anti-p63, anti-LSR, anti-OCLN, anti-CGN, anti-ZO-3, anti-TRIC, anti-CLDN-4, anti-acetylated-tubulin (1:100), and anti-cytokeratin 5 (1:200) antibodies and Alexa 594-phalloidin (1:200) overnight at 4 °C. Alexa Fluor 488 (green)-conjugated anti-rabbit IgG and Alexa Fluor 594 (red)-conjugated anti-mouse IgG (Invitrogen) were used as secondary antibodies. The specimens were examined and photographed with an Olympus IX 71 inverted microscope (Olympus Co.; Tokyo, Japan) and a confocal laser scanning microscope (LSM510; Carl Zeiss, Jena, Germany).

### 2.11. Cell Cycle Assay

A253 cells cultured in 35 mm dishes were collected with 0.05% Trypsin-EDTA and washed once with PBS. After that, the cells were added to 1 ml of ice cold 70% ethanol and incubated for at least 3 h at −20 °C. The cells were then washed once with PBS and with 200 μL of Muse Cell Cycle reagent (Merck Millipore, MA, USA), before being incubated for 30 min at room temperature in the dark. We used a Muse® Cell Analyzer to measure the cell cycle according to the manufacturer’s instructions.

### 2.12. Migration Assay

After the A253 cells were plated onto 35 mm dishes, they were cultured to confluence. At 24 h, we wounded the cell layer using a plastic pipette tip (P200), and then measured the length of the wound by using a microscope imaging system (Olympus, Tokyo, Japan).

### 2.13. Measurement of Transepithelial Electrical Resistance (TEER)

A253 cells were cultured to confluence in the inner chambers of 12 mm transwell inserts with 0.4 μM pore-size filters (Corning Life Sciences). The TEER was measured using an EVOM voltameter with an ENDOHM-12 (World Precision Instruments, Sarasota, FL, USA) on a heating plate (Fine, Tokyo, Japan) adjusted to 37 °C. The values were expressed in standard units of ohms per square centimeter and presented as the mean ± S.D. For calculation, the resistance of blank filters was subtracted from that of filters covered with cells.

### 2.14. XF96 Extracellular Flux Measurements

Mitochondrial respiration was assessed using an XF96 Extracellular Flux Analyzer (Aligent, Santa Clara, CA, USA). A253 cells were seeded on XF96 plates at a density of 20,000 cells/well after incubation in DMEM medium with a high glucose or glucose -free medium for 24 h. A day prior to the experiment, sensor cartridges were hydrated with XF calibrate solution (pH 7.4) and incubated at 37 °C in a non-CO_2_ incubator for 24 h. Baseline measurements of mitochondrial respiration (OCR) were taken before sequential injection of the following inhibitors: 1 μM oligomycin, which is an ATP synthase inhibitor; 2 μM FCCP, which is a mitochondrial respiration uncoupler; and 1 μM antimycin A and rotenone, which are mitochondrial electron transport blockers. Oligomycin was applied first to estimate the proportion of basal OCR coupled to ATP synthesis. After oligomycin application, FCCP was used to further determine the maximal glycolysis pathway capacity.

### 2.15. Data Analysis

Each set of results shown is representative of at least three separate experiments. Results are given as means ± SEM. Statistical analysis was by one-way analysis of variance (ANOVA), followed by a post-hoc test and an unpaired two-tailed Student’s *t*-test. Statistical significance was set at * *p* < 0.05 and ** *p* < 0.01.

## 3. Results

### 3.1. Knockdown of p63 Induced Tight Junction (TJ) Proteins CGN and ZO-3 and Differentiation Markers in Human Salivary Gland Duct Adenocarcinoma Cell Line A253

p63 regulated via TGF-β/JNK signaling, is a diagnostic marker of salivary gland neoplasia [13,17]. To investigate the roles of p63 expression in human salivary gland duct adenocarcinoma, human salivary gland duct adenocarcinoma cell line A253 was transfected with siRNA-p63 in the presence of a TGF-β receptor type 1 inhibitor (EW-7197) and JNK inhibitor (SP600125), before GeneChip analysis was performed on the cells. In the treated cells, downregulation of p63 and upregulation of tight junction proteins CGN and ZO-3 were observed compared to the control (Table 1). Furthermore, the epithelial differentiation markers OLFM3, CY1A1, AQP7, TLR1, TLR5, and ELF3 were markedly increased (Table 1).

### 3.2. Knockdown of p63 Induced Tight Junction (TJ) Proteins at the Membranes in A253 Cells

As upregulation of mRNAs of tight junction (TJ) proteins CGN and ZO-3 was observed by knockdown of p63 in GeneChip analysis. We first investigated the expression and localization of p63 and TJ proteins in A253 cells. Immunocytochemical analysis revealed that p63 was expressed in the nuclei of all cells and that CGN was weakly expressed at the membranes, whereas expression of angulin-1/LSR, OCLN, and ZO-3 was not detected at the membranes. When A253 cells were transfected with siRNA of p63 in the presence of EW-7197 and SP600125, p63 disappeared from the nuclei, angulin-1/LSR, OCLN, and ZO-3 were recruited at the membranes, and CGN at the membranes was increased (Figure 1A). In the Western blot analysis, knockdown of p63 by the siRNA-p63 with or without EW-7197 and SP600125 did not affect expression of the TJ proteins (Figure 3A,B).

### 3.3. HDAC Inhibitors Downregulate Expression of p63 and Upregulate Expression of TJ Proteins in A253 Cells

HDAC inhibitors downregulate p63 expression [12]. To investigate whether HDAC inhibitors affect expression of p63 and the TJ proteins, A253 cells were treated with the HDAC inhibitors trichostatin A (TSA) and quisinostat (JNJ -26481585) in the presence of EW-7197 and SP600125. Immunocytochemical analysis showed that p63 disappeared from the nuclei and angulin-1/LSR, OCLN, and ZO-3 were recruited at the membranes, and CGN, TRIC, and CLDN-4 at the membranes were increased by treatment with TSA and quisinostat in the presence of EW-7197 and SP600125. In the control, TRIC and CLDN-4 were weakly detected at the membranes (Figure 1A).

In TEM analysis, membrane adhesion between the cells and clear TJ structures was observed in A253 cells treated with TSA and quisinostat in the presence of EW-7197 and SP600125, while in the control and the treatment with only EW-7197 and SP600125, spaces between the cells were observed and clear TJ structures were not found (Figure 2A). In the Western blot analysis, downregulation of p63 was observed after transfection with siRNA-p63 and treatment with TSA and quisinostat with or without EW-7197 and SP600125, while treatment with TSA and quisinostat with or without EW-7197 and SP600125 induced expression of CGN and acetylated tubulin (Figure 3A,B). However, no change of ZO-3, OLFML3, CYP1A1, angulin-1/LSR, TRIC, CLDN-4, and CLDN-7 proteins were observed (Figure 3A,B). Immunocytochemical analysis showed that expression of CGN and ZO-3 recruited at the membranes increased and changed from dots to lines in transfection with siRNA-p63 and treatment with TSA and quisinostat with or without EW-7197 and SP600125.

Furthermore, we investigated the effects of HDAC inhibitors in the presence of EW-7197 and SP600125 on epithelial barrier function in A253 cells (Figure 2B). Treatment with the HDAC inhibitors TSA and quisinostat at both 1 and 10 μM in the presence of EW-7197 and SP600125 increased the epithelial barrier function, indicated as TEER values (Figure 2B).

### 3.4. NF-κB Inhibitor Curcumin Downregulate Expression of p63 and Upregulate Expression of TJ Proteins in A253 Cells

p63 may be regulated by NF-κB [5]. To investigate the effects of p63 via a NF-κB signal pathway on TJ proteins in SDC, A253 cells were treated with the NF-κB inhibitor curcumin at 10 μM under pretreatment with TGF-β type I receptor inhibitor EW-7197 and JNK inhibitor SP600125 at 10 μM. In immunocytochemical analysis, p63 disappeared from the nuclei, and angulin-1/LSR, OCLN, TRIC, and CLDN-4 were recruited at the membranes by treatment with curcumin under pretreatment, with or without EW-7197 and SP600125 (Figure 4A). In Western blot analysis, downregulation of p63, phosphorylated NF-κB, and phosphorylated MAPK, and upregulation of CLDN-4, were observed by treatment with curcumin under pretreatment, with or without EW-7197 and SP600125 (Figure 4B).

### 3.5. CGN Induced by HDAC Inhibitors Binds Microtubules Formed Planar Apical Network in A253 Cells

CGN is a TJ protein that binds to both actin filaments and microtubules (MTs), and which localizes to centrosomes [51,52]. Furthermore, the planar apical network of non-centrosomal MTs associated with TJs including CGN and the CGN may organize the apical MT network’s formation [51]. Therefore, we investigated the detailed relationship between CGN and MTs induced by HDAC inhibitors in A253 cells.

We performed coimmunoprecipitation using a CGN antibody for A253 cells treated with TSA and quisinostat in the presence of EW-7197 and SP600125. In the CGN-immunoprecipitate of the treated cells, acetylated-tubulin was clearly detected (Figure 3D), while TRIC, but not LSR and CLDN-7, was weakly detected (Appendix A). In immunocytochemical analysis, in the treated cells which expressed CGN at the membranes, the apical MT network’s formation indicated by α-tubulin was observed, while in the control, clear formation was not observed (Figure 3E).

### 3.6. Knockdown of p63 and HDAC Inhibitors Prevent Cell Proliferation in A253 Cells

p63 contributes to cell proliferation, and HDAC inhibitors downregulated p63 expression, preventing cell proliferation in cancer cells [5,12]. We investigated the effects of the knockdown of p63 and HDAC inhibitors in the presence of EW-7197 and SP600125 on cell proliferation of A253 cells. In the cell cycle assay, the G0/G1 phase was significantly increased, and the G2/M phase was decreased by the knockdown of p63 by the siRNA-p63 (Figure 5A). In the treatment with TSA at 1 μM in the presence of EW-7197 and SP600125, the G0/G1 phase was significantly increased, and the G2/M phase was decreased, and in treatment with TSA at 10 μM in the presence of EW-7197 and SP600125, the G2/M phase was significantly decreased (Figure 5B). In treatment with quisinostat at both 1 and 10 μM in the presence of EW-7197 and SP600125, the G0/G1 phase was significantly increased, and the G2/M phase was decreased (Figure 5C).

### 3.7. HDAC Inhibitors Prevent Cell Migration in A253 Cells

HDAC inhibitors prevent cell migration in cancer cells [23]. We investigated the effects of HDAC inhibitors in the presence of EW-7197 and SP600125 on cell migration. In cell migration assay, both HDAC inhibitors in the presence of EW prevented cell migration, while treatment with only EW-7197 and SP600125 also prevented cell migration (Figure 5D,E).

### 3.8. Knockdown of p63 Induced Aberrant Cell Metabolism in A253 Cells

The reprogramming of cell metabolism is one of the hallmarks of cancers [54]. We investigated the effects of knockdown of p63 by the siRNA of p63 on cell metabolism in A253 cells. In mitochondrial stress tests using Seahorse Bioscience XF Analyzers, the knockdown of p63 by siRNA of p63 decreased the mitochondrial respiration, measured by the baseline oxygen consumption rate (OCR), proton leak, maximal respiration, spare respiratory capacity, non-mitochondrial oxygen consumption, ATP production, coupling efficiency, and spare respiratory capacity, compared with the control (Figure 5F–H). These results indicated that the knockdown of p63 prevented cell metabolism in salivary duct adenocarcinoma A253 cells. We also investigated the effects of HDAC inhibitors in the presence of EW-7197 and SP600125 on cell metabolism in A253 cells. In the treatment with both HDAC inhibitors TSA and quisinostat in the presence of EW-7197 and SP600125, no change of OCR was observed compared to the control (Appendix A).

### 3.9. Roles of p63, CGN and Angulin-1/LSR in Cultured Normal Human Salivary Gland Duct Epithelial (HSDE) Cells

p63 maintains cell proliferation of p63+/CK5+ stem/progenitor cells in salivary gland [18]. To examine the roles of p63, CGN and angulin-1/LSR in normal human salivary gland duct epithelial cells, we used HSDE cells derived from salivary gland tissues of patients with IgG4-related disease that were previously reported [55]. In immunocytochemical analysis, p63 in the nuclei and CK5 in the cytoplasm were detected in the cultured normal HSDE cells (Figure 6A). When HSDE cells were transfected with siRNA of p63, p63 disappeared and OCLN was recruited at the membranes, while OCLN was not detected at the membrane in the control (Figure 6A). In Western blot analysis, the knockdown of p63 decreased the expression of angulin-1/LSR, TRIC, CLDN-1, and CLDN-7, and increased the expression of CLDN-4 in HSDE cells (Figure 6B). When HSDE cells were transfected with siRNA of CGN, the knockdown of CGN decreased the expression of ZO-3, angulin-1/LSR, TRIC, CLDN-1, pMAPK, OLFML3, and CYP1A1, and increased the expression of CLDN-4 (Figure 5B). When HSDE cells were transfected with siRNA of angulin-1/LSR, knockdown of LSR decreased the expression of ZO-3, CLDN-1, OLFML3, and CYP1A1, and increased the expression of TRIC, CGN, and CLDN-4 (Figure 6B).

### 3.10. Expression and Localization of p63, CGN, ZO-3 and OLFML3 in Human Salivary Duct Adenocarcinoma

P63/p40 (ΔNp63) is used for diagnostic utility in the histologic differentiation of salivary gland tumors [13]. We investigated the expression and localization of p63, CGN, ZO-3, and OLFML3 in human salivary duct adenocarcinoma, using paraffin-embedded sections of salivary duct adenocarcinoma tissues. In immunohistochemical analysis, CGN and ZO-3 at the membrane and OLFML3 in the cytoplasm were detected in peripheral salivary gland duct epithelial cells of normal salivary gland tissues, while p63 was detected in the myoepithelial cells (Figure 7). In salivary gland duct adenocarcinoma gland-like structures, the expression of CGN and ZO-3, but not p63, was detected, and OLFML3 was decreased in the cancer cells compared to normal ones, while in the solid type of salivary gland duct adenocarcinoma, CGN, ZO-3, and OLFML3 were not detected, while p63 was detected in some cancer cells (Figure 7).

## 4. Discussion

Inhibition of HDAC and signal transduction pathways promoted differentiation of p63-positive SDC A-253 cells. HDAC inhibitors TSA and quisinostat, under pretreatment with TGF-β type I receptor inhibitor EW-7197 and JNK inhibitor SP600125, promoted differentiation of p63-positive salivary gland duct adenocarcinoma cells and, as a result, suppressed cell proliferation and migration, and increased the epithelial barrier function with recruitment of the TJ proteins CGN, ZO-3, angulin-1/LSR, and OCLN in salivary gland duct adenocarcinoma.

p63, which has two distinct isoforms, proapoptotic TAp63 and oncogenic ΔNp63, is an essential gene for the proliferation and differentiation of epithelial cells [5,6]. ΔNp63 can also regulate the expression of genes correlated with EMT [10]. It has been reported that p63 regulates several TJ proteins in cancer cells [56,57]. Furthermore, it is known that p63 regulates the expression of the TJ protein JAM-A, which correlates with poor prognosis in various cancers [57,58,59,60]. In ESCC cell lines, the silencing of p63 mRNA increases expression of components of tight junctions and downregulates cell proliferation [61].

In this study, a DNA array showed that the knockdown of p63, under pretreatment with TGF-β type I receptor inhibitor EW-7197 and JNK inhibitor SP600125, downregulated p63 mRNA and upregulated mRNAs of the TJ proteins CGN and ZO-3, as well as the epithelial differentiation markers OLFM3, CY1A1, AQP7, TLR1, TLR5, and ELF3. The knockdown of p63 under pretreatment with EW-7197 and SP600125, recruited the TJ proteins CGN, ZO-3, angulin-1/LSR, and OCLN at the membranes and prevented cell proliferation. The cell differentiation of salivary gland duct carcinoma cells may be accompanied by a decrease in p63, and by a shift to adhesion patterns involving tight junctions. Furthermore, treatment with NF-κB inhibitor curcumin under pretreatment, with or without EW-7197 and SP600125, downregulated p63, upregulated CLDN-4, and recruited the TJ proteins angulin-1/LSR, OCLN, TRIC, and CLDN-4 at the membranes.

On the other hand, histone deacetylases (HDACs), which affect the DNA structure and which also able to regulate its transcription, repair, and replication, are thought to play a crucial role in carcinogenesis. It is understood that HDAC inhibitors are epigenetic regulators that relax chromatin structures through increased histone acetylation, and consequently promote the expression of repressed genes. Currently, several HDAC inhibitors are in clinical use as anticancer agents. In salivary gland cancer, HDAC inhibitors combined with epithelial growth factor receptor (EGFR) inhibitors exert synergistic and potent cytotoxic effects [30]. We previously reported that the HDAC inhibitors TSA and quisinostat suppressed malignancy via changes of the expression of TJ proteins in lung adenocarcinoma, and that TSA also suppressed the proliferation, migration, and invasion of HNSCC cells via p63-mediated tight junction molecules and p21-mediated growth arrest [62,63]. In the present study, the HDAC inhibitors TSA and quisinostat, under pretreatment with the TGF-β type I receptor inhibitor EW-7197 and JNK inhibitor SP600125, downregulated expression of p63 and upregulated expression of TJ proteins with the epithelial barrier, as well as preventing cell proliferation and cell migration. In the present study, HDAC inhibitors strongly induced protein expression of CGN, but not ZO-3, in Western blotting analysis, while the knockdown of p63 resulted in the recruitment of CGN and ZO-3 at the membrane in the immunofluorescence analysis. In the effects of HDAC inhibitors for TJ proteins, there is not only a p63 pathway, but also other mechanisms, including the NF-κB signal pathway.

In addition, CGN is a TJ protein that binds to both actin filaments and microtubules (MTs), and localizes to centrosomes [51,52]. Furthermore, a planar apical network of non-centrosomal MTs is associated with TJs, including CGN, and the CGN may organize the apical MT network’s formation [51]. In the present study, in the CGN-immunoprecipitate of the cells treated with TSA and quisinostat in the presence of EW-7197 and SP600125, CGN and acetylated-tubulin were detected. In immunocytochemical analysis of the treated cells that expressed CGN at the membranes, the formation of the apical MT network, indicated by α-tubulin, was observed. It was thought that the network formation induced by the presence of CGN indicated epithelial differentiation. Although CGN and ZO-3 may be regulated via various signaling pathways, the mechanisms remain unclear. In the present study, TGF-β type I receptor inhibitor EW-7197 and JNK inhibitor SP600125 enhanced expression of GCN and ZO-3 induced by transfection with siRNA of p63 and treatment with TSA and JNJ at the membranes. These findings indicated that CGN and ZO-3 were in part regulated via TGF-β/JNK signaling and that the signaling pathways might play crucial roles in the differentiation of salivary gland duct epithelial cells.

Meanwhile, the Warburg effect is metabolic reprogramming, which indicates metabolic switching in cancer cells from oxidative phosphorylation to aerobic glycolysis by decreasing the mitochondrial respiration [64]. Reprograming of cellular metabolism is one of the keys to cancer research and treatment [65]. Accordingly, we investigated whether p63 and HDAC affected cellular metabolism in salivary gland duct adenocarcinoma by using mitochondrial stress tests. The knockdown of p63 decreased mitochondrial respiraition, but HDAC inhibitors did not affect it. TAp63d participates in myoblasts metabolism control [66]. These results indicate that p63 also plays an important role in cellular metabolism in salivary gland duct adenocarcinoma.

As the data reported here, p63 plays a crucial role in salivary adenocarcinoma in vitro, whereas p63/p40 (ΔNp63) is used for diagnostic utility in the histologic differentiation of salivary gland tumors [13]. In the present study, in salivary gland duct adenocarcinoma gland-like structures, the expression of CGN and ZO-3, but not p63, was detected, and OLFML3 was decreased in the cancer cells compared to normal ones, while in the solid type of salivary gland duct adenocarcinoma, CGN, ZO-3, and OLFML3 were not detected, while p63 was detected in some cancer cells.

Conversely, the knockout of ΔNp63 results in a loss of the stem/progenitor cell population in the salivary gland and promotes the differentiation mediated by dysregulated TGF-β/activin signaling [18]. p63 induces keratinocyte differentiation via the c-Jun N-terminal kinase pathway [67]. In normal human salivary gland duct epithelial (HSDE) cells in which p63 was positive, the knockdown of p63 recruited OCLN at the membrane, and downregulation of CLDN-1 and upregulation of CLDN-4 were observed after the knockdown. The knockdown of CGN downregulated expression of TRIC, CLDN-1, OLFML3, and CY1A1, and upregulated CLDN-4 expression. The knockdown of angulin-1/LSR upregulated expression of CGN, TRIC, and CLDN-4, and downregulated ZO-3 and CLDN-1. Although it is necessary to perform more experiments, it is thought that in p63-positive HSDE cells the mechanisms regulated by p63 may be different from salivary duct carcinoma. Furthermore, CGN and angulin-1/LSR may play crucial roles in the tight junctions of HSDE cells.

## 5. Conclusions

P63 is not only a diagnostic marker of salivary gland neoplasia, but it also promotes malignancy. Inhibition of HDAC and signal transduction pathways inhibited cell proliferation and migration, induced tight junctions, and promoted differentiation in p63-positive salivary duct adenocarcinoma (SDC). Inhibition of HDAC and signal transduction pathways is useful in therapy for p63-positive SDC cells.

## Figures and Tables

**Figure 1 cancers-14-02584-f001:**
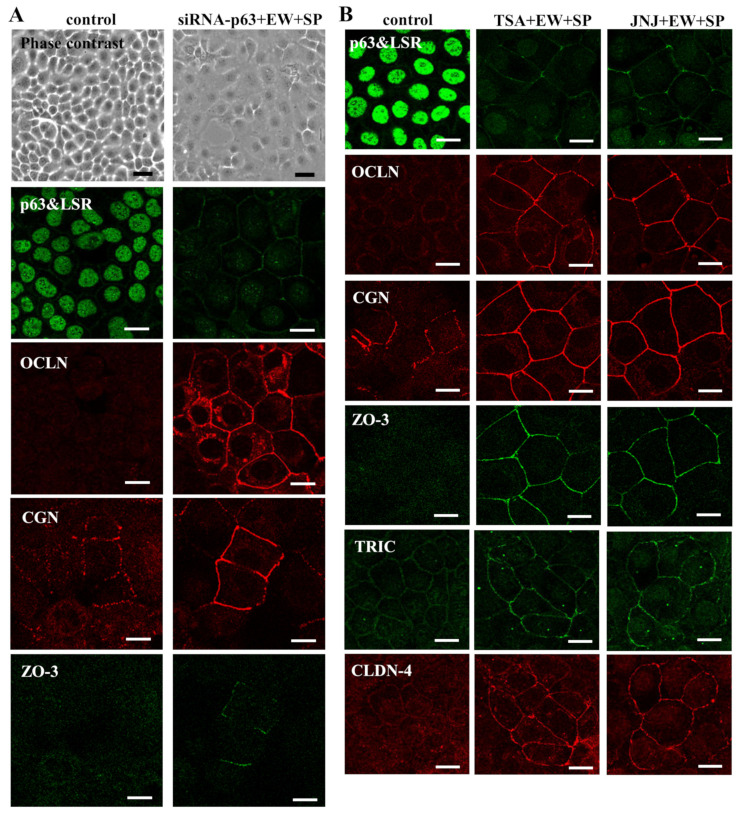
(**A**) Phase-contrast and immunocytochemical staining for p63, angulin-1/LSR, OCLN, CGN, and ZO-3 in A253 cells transfected with siRNA-p63 in the presence of EW-7197 and SP600125 at 10 μM. Scale black bar: 40 μM, white bar: 10 μM. (**B**) Immunocytochemical staining for p63, angulin-1/LSR, OCLN, CGN, ZO-3, TRIC, and CLDN-4 in A253 cells treated with TSA and JNJ at 10 μM in the presence of EW-7197 and SP600125 at 10 μM. Scale bar: 10 μM.

**Figure 2 cancers-14-02584-f002:**
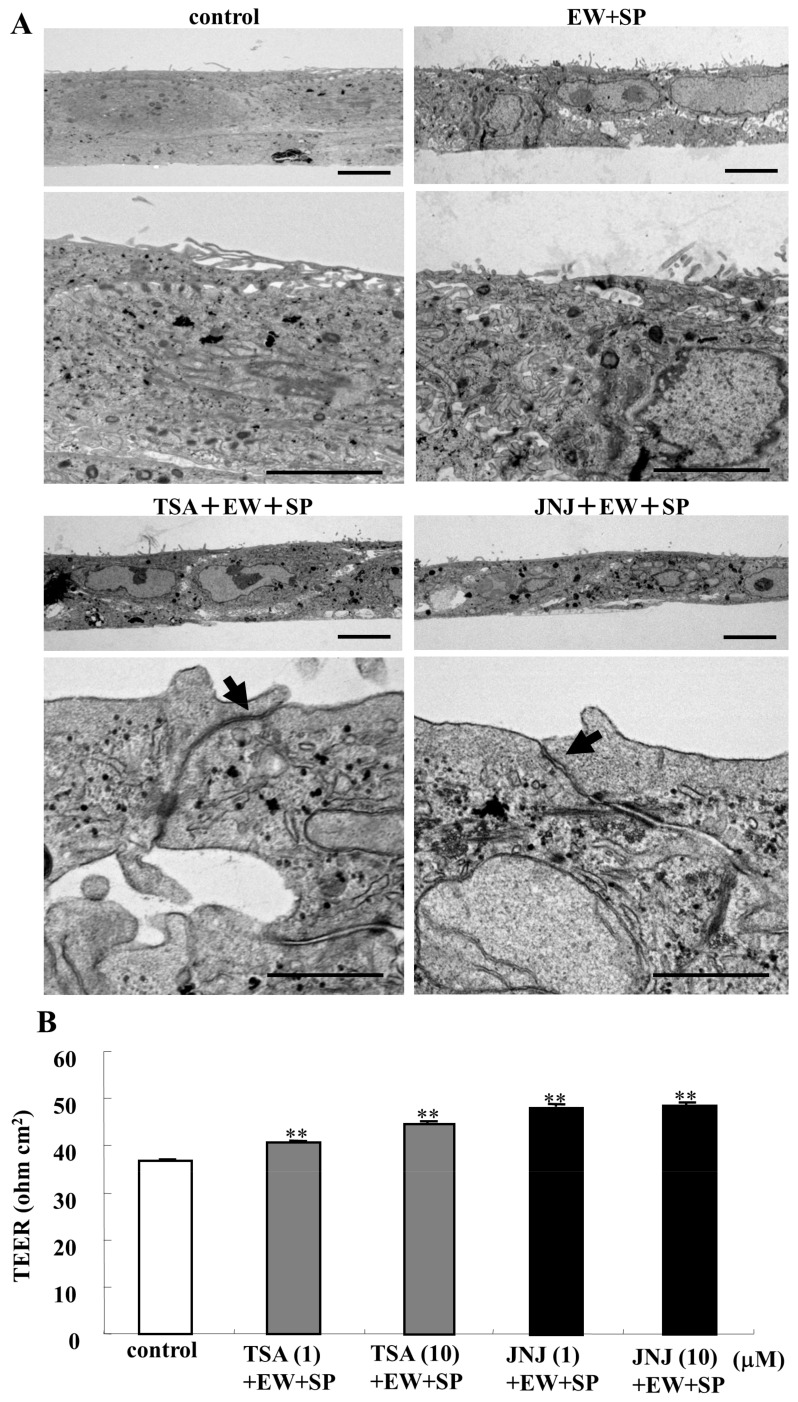
(**A**) Transmission electron microscopic (TEM) analysis of A253 cells treated with or without TSA and JNJ at 10 μM in the presence of EW-7197 and SP600125 at 10 μM Scale bar: 2 μM. (**B**) Bar graph TEER values representing barrier function in A253 cells treated with TSA and JNJ at 1 and 10 μM in the presence of EW-7197 and SP600125 at 10 μM. *p* ** <0.01.

**Figure 3 cancers-14-02584-f003:**
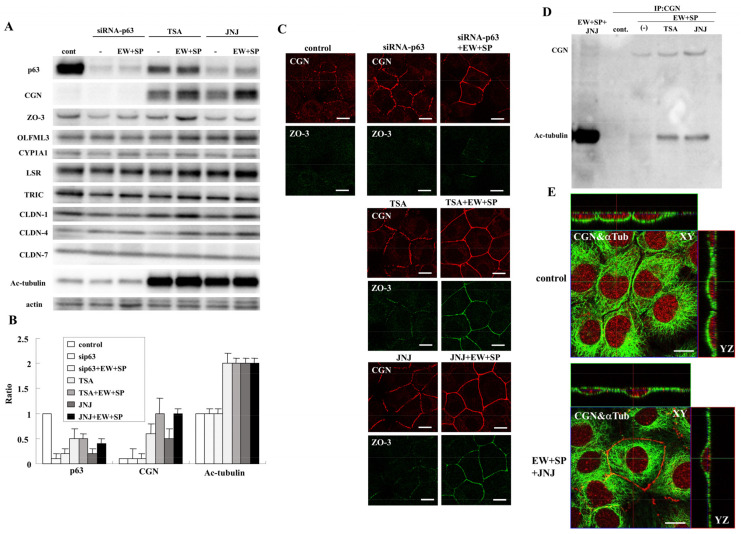
(**A**) Western blotting for p63, CGN, ZO-3, OLFML3, CY1A1, angulin-1/LSR, TRIC, CLDN-1, CLDN-4, CLDN-7, Ac-tubulin, and actin in A253 cells transfected with siRNA-p63 and treated with TSA and JNJ at 10 μM with or without EW-7197 and SP600125 at 10 μM. (**B**) The results of (**A**) are shown as a bar graph. (**C**) Immunocytochemical staining for CGN and ZO-3 in A253 cells transfected with siRNA-p63 and treated with TSA and JNJ at 10 μM with or without EW-7197 and SP600125 at 10 μM. Scale bar: 10 μM. (**D**) Coimmunoprecipitation in A253 cells treated with or without TSA and JNJ at 10 μM in the presence of EW-7197 and SP600125 at 10 μM. Immunoprecipitation using anti-CGN antibody led to the identification of Ac-tubulin in Western blot analysis. (**E**) Double-immunocytochemical staining for CGN and α-tubulin in A253 cells treated with JNJ at 10 μM in the presence of EW-7197 and SP600125 at 10 μM. Scale bar: 10 μM.

**Figure 4 cancers-14-02584-f004:**
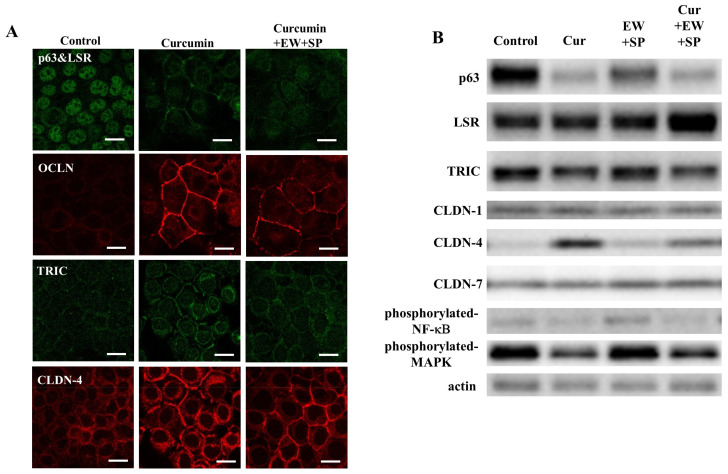
(**A**) Images of immunocytochemical staining for p63, LSR, OCLN, TRIC, and CLDN-4 in A253 cells treatment with NF-κB inhibitor curcumin at 10 μM under pretreatment, with or without EW-7197 (EW) and SP600125 (SP) at 10 μM. Scale bar: 20 µM. (**B**) Western blotting for p63, angulin-1/LSR, TRIC, CLDN-1, CLDN-4, CLDN-7, pNF-κB, pMAPK, and actin in A253 cells treated with curcumin (Cur) at 10 μM under pretreatment, with or without EW-7197 (EW) and SP600125 (SP) at 10 μM.

**Figure 5 cancers-14-02584-f005:**
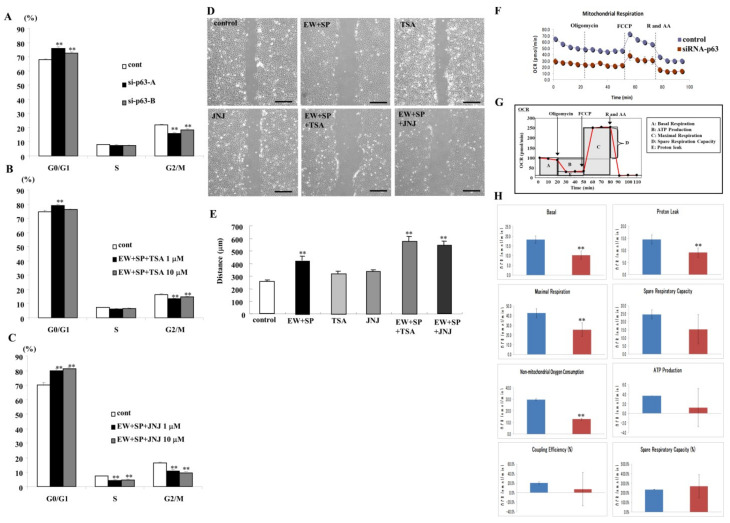
Cell cycle assay of A253 cells transfected with siRNA-p63 (**A**) and treated with or without TSA (**B**) and JNJ (**C**) at 1 and 10 μM in the presence of EW-7197 and SP600125 at 10 μM. ** *p* < 0.01. (**D**) Images of scratch wound assay of A253 cells treated with or without TSA and JNJ at 10 μM with or without EW-7197 and SP600125 at 10 μM. Scale bar: 200 μM. (**E**) The result of (**D**) is shown as a bar graph. (**F**) Line graph and (**H**) bar graphs of OCR in A253 cells transfected with siRNA of p63. (**G**) Schema of OCR measured the key parameters of mitochondria function in line graph. Steady-state OCR was measured at six time points. Oligomycin was injected to inhibit ATP synthase, with the addition of FCCP to uncouple mitochondria and obtain the maximal oxygen consumption rate at the eighth time point. Finally, rotenone and antimycin A (R and AA) were injected to confirm that the respiration changes were due mainly to mitochondrial respiration. ** *p* < 0.01.

**Figure 6 cancers-14-02584-f006:**
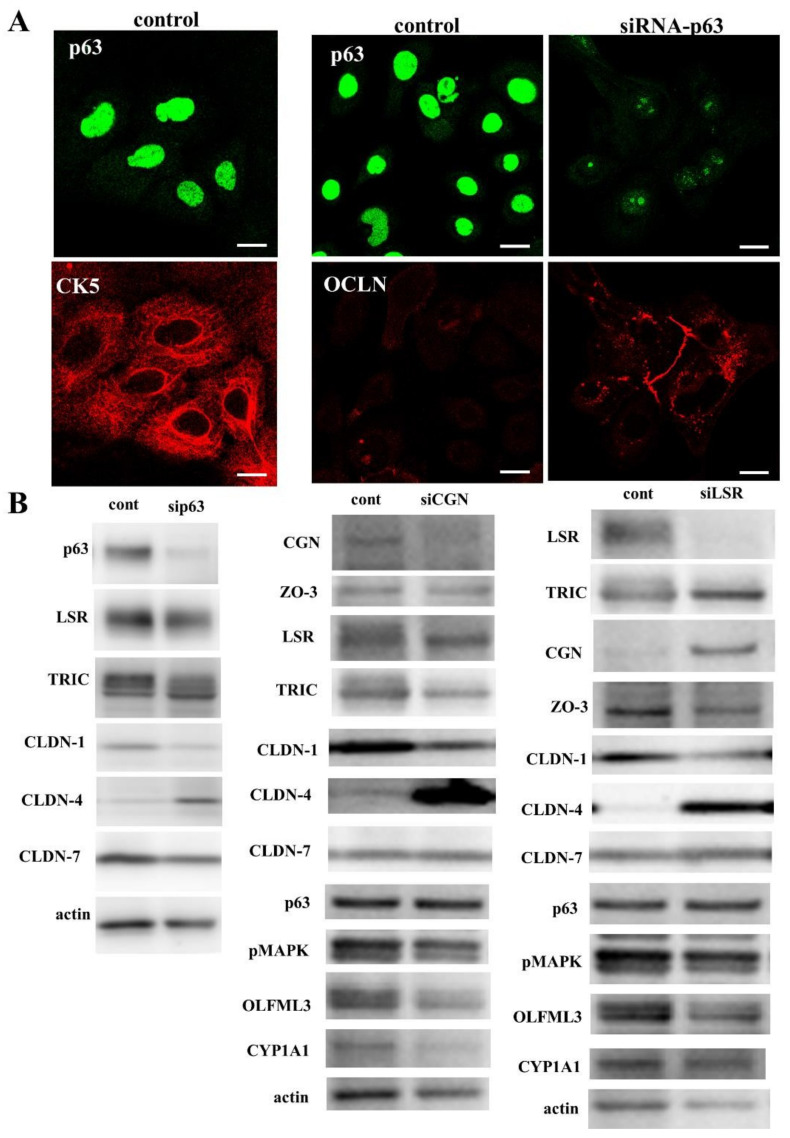
(**A**) Immunocytochemical staining for p63, CK5, and OCLN in cultured normal human salivary gland duct epithelial (HSDE) cells transfected with or without siRNA-p63. Scale bar: 10 μM. (**B**) Western blotting for p63, CGN, ZO-3, angulin-1/LSR, TRIC, CLDN-1, CLDN-4, CLDN-7, pMAPK, OLFML3, CY1A1, and actin in A253 cells transfected with siRNA-p63, CGN or angulin-1/LSR.

**Figure 7 cancers-14-02584-f007:**
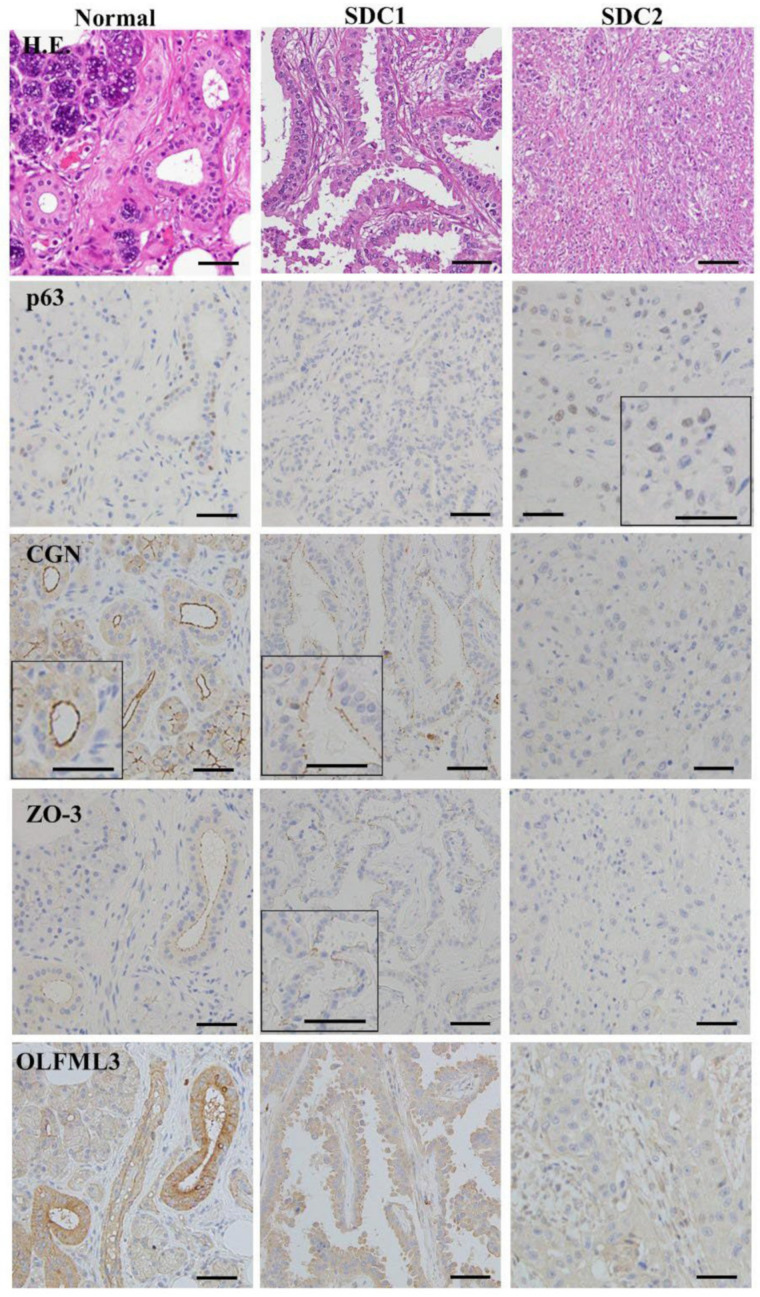
Images of H.E. and immunohistochemical staining of p63, CGN, ZO-3, and OLFML3 in tissues of human salivary duct adenocarcinoma patients (SDC1, SDC2). Scale bar: 200 μM.

**Table 1 cancers-14-02584-t001:** List of gene probes up- or downregulated in A253 cells transfected with siRNA-p63 in the presence of EW-7197 and SP600125.

Gene Name	ID	Gene Bank ID	Fold-Change Control vs. Treatment
TP63	H300019931	NM_001114982.1	0.34
CGN	H30009163	NM_020770.2	2.77
TJP3 (ZO-3)	H200003536	NM_001267561.1	12.47
OLFL3	AHsV10002740	NM_020190.3	14.56
CY1A1	AHsV10002241	NM_001319217.1	66.44
AQP7	H300004487	XM_011517868.1	2.26
TLR1	CHsGV10003913	NM_003263.3	2.38
TLR5	H200011920	NM_003268.5	2.28
ELF3	H200007656	NM_001114309.1	2.36

## Data Availability

Data is contained within the article.

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
