# Peer review of "Inhibition of HDAC and Signal Transduction Pathways Induces Tight Junctions and Promotes Differentiation in p63-Positive Salivary Duct Adenocarcinoma"

_cancers, 2022, doi:10.3390/cancers14112584_

Round 1

Reviewer 1 Report

In this manuscript Nakano and colleagues aimed at describing the role of p63 and HDAC inhibitors in the differentiation and migration of salivary duct adenocarcinoma cell lines. Although authors state in the abstract and in Result and Discussion sections, that knock-down of p63 increases the expression of TJ proteins and decreases cell proliferation, in all the experiments that they have performed they also treated the cell lines with two inhibitors (the TGF-βR1 inhibitor EW-7197 and the JNK inhibitor SP600125) that authors have already shown to have effect in the expression of TJ proteins and p63 (Effects of HMGB1 on Tricellular Tight Junctions via TGF-Signaling in Human Nasal Epithelial Cells and Epithelial barrier dysfunction and cell migration induction via JNK/cofilin/actin by angubindin-1). Even though authors conclude that the results obtained are due to p63 kd, they lack sufficient controls (EW-7197 and SP600125 treatment w/o sip63) to give this conclusion. Moreover, in the one experiment that they performed knocking-down p63, and looking at the expression of TJ proteins without EW-7197 and SP600125 treatment (Figure 3A), lack of p63 does not increase TJ proteins expression, ZO-3 expression even decreases upon p63 silencing, which is the opposite of what they showed in previous results (table 1 and Figure 1A). Same thing using HDAC inhibitors, since in most of the experiments TSA and JNJ were used in addition to EW-7197 and SP600125, authors cannot conclude that the effects that they see are due to  HDAC inhibition. Moreover, in the only experiments where authors showed the effect of TSA and JNJ alone or in combination with EW-7197 and SP600125 (Figure 4E), it is clear that EW-7197 and SP600125 treatment gives a better response compared to TSA or JNJ alone. In addition, authors also state that HDACi decrease the expression of p63 (3.3. HDAC inhibitors downregulate expression of p63 and upregulate expression of TJ proteins in A253 cells) (Figure 1B), but also this experiment was performed in combination with EW-7197, drug that authors have already shown to decrease p63 expression and TJ markers (Effects of HMGB1 on Tricellular Tight Junctions via TGF-Signaling in Human Nasal Epithelial Cells). Moreover, in figure 3C authors claimed the interaction between CGN and ac-tubulin upon TSA and EW treatment, even in this experiments authors should have shown the binding to tubulin and not only ac-tubulin, which is known to increase upon TSA treatment.

In addition, also the experiments are poorly described, in the immunofluorescence analysis shown in Figure 1A and 1B, it seems that authors co-stained the cells with p63 and LSR antibody, is that an error?

Finally, despite authors have started their work with the hypothesis, already published (p63 expression in human tumors and normal tissues: a tissue microarray study on10,200 tumors) that p63 is overexpressed in salivary gland carcinoma, they couldn’t even confirm this result in their system, something that do not support at all the importance of p63 in this study.

Considering that it is very difficult to understand the design of the experiments and in many cases their meaning, that authors didn’t even discuss in a proper manner their results and present a very poor discussion, I think that this manuscript presents too many lacks to be published.  

Author Response

Response to Comments and Suggestions of Reviewer 1

In this manuscript Nakano and colleagues aimed at describing the role of p63 and HDAC inhibitors in the differentiation and migration of salivary duct adenocarcinoma cell lines. Although authors state in the abstract and in Result and Discussion sections, that knock-down of p63 increases the expression of TJ proteins and decreases cell proliferation, in all the experiments that they have performed they also treated the cell lines with two inhibitors (the TGF-βR1 inhibitor EW-7197 and the JNK inhibitor SP600125) that authors have already shown to have effect in the expression of TJ proteins and p63 (Effects of HMGB1 on Tricellular Tight Junctions via TGF-Signaling in Human Nasal Epithelial Cells and Epithelial barrier dysfunction and cell migration induction via JNK/cofilin/actin by angubindin-1). Even though authors conclude that the results obtained are due to p63 kd, they lack sufficient controls (EW-7197 and SP600125 treatment w/o sip63) to give this conclusion.

As you know, we previously reported the mechanisms in regulation of TJ proteins including tricellular TJ protein angulin-1/LSR via p63 and the effects of TGF-βR1 inhibitor EW-7197 and the JNK inhibitor SP600125 in normal human epithelial cells. In the present study, we indicated the mechanisms in regulation of TJ proteins including tricellular TJ protein angulin-1/LSR via p63 and the effects of TGF-βR1 inhibitor EW-7197 and the JNK inhibitor SP600125 in p63-posive cancer cells. Accordingly, it is thought that the findings in the present study are novel and useful in cancer therapy.

Moreover, in the one experiment that they performed knocking-down p63, and looking at the expression of TJ proteins without EW-7197 and SP600125 treatment (Figure 3A), lack of p63 does not increase TJ proteins expression, ZO-3 expression even decreases upon p63 silencing, which is the opposite of what they showed in previous results (table 1 and Figure 1A). Same thing using HDAC inhibitors, since in most of the experiments TSA and JNJ were used in addition to EW-7197 and SP600125, authors cannot conclude that the effects that they see are due to HDAC inhibition.

We also expected that knockdown of p63 might induced expression of CGN and ZO-3 proteins in Western blotting analysis like the results of GeneChip analysis. However, the knockdown of p63 resulted in the recruitment of CGN and ZO-3 at the membrane in the immunofluorescence analysis. In the present study, the mechanisms remain unclear. On the other hand, HDAC inhibitors strongly induced protein expression of CGN but not ZO-3 in Western blotting analysis. We think that in the effects of HDAC inhibitors, there are not only p63 pathway but also another mechanisms. We added the sentence in Discussion.

Moreover, in the only experiments where authors showed the effect of TSA and JNJ alone or in combination with EW-7197 and SP600125 (Figure 4E), it is clear that EW-7197 and SP600125 treatment gives a better response compared to TSA or JNJ alone. In addition, authors also state that HDACi decrease the expression of p63 (3.3. HDAC inhibitors downregulate expression of p63 and upregulate expression of TJ proteins in A253 cells) (Figure 1B), but also this experiment was performed in combination with EW-7197, drug that authors have already shown to decrease p63 expression and TJ markers (Effects of HMGB1 on Tricellular Tight Junctions via TGF-Signaling in Human Nasal Epithelial Cells).

We agree with the comments. We added the data of the immunofluorescence analysis with or without EW-7197 and SP600125 as Figure 3C. Expression of CGN and ZO-3 recruited at the membranes increased and changed from dots to lines in transfection with siRNA-p63 and treatment with TSA and quisinostat with EW-7197 and SP600125 than without EW-7197 and SP600125. We added the sentence in Results.

Moreover, in figure 3D authors claimed the interaction between CGN and ac-tubulin upon TSA and EW treatment, even in this experiments authors should have shown the binding to tubulin and not only ac-tubulin, which is known to increase upon TSA treatment.

HDAC inhibitors TSA and JNJ induce acetylation in various proteins. It is known that CGN binds to tubulin and organizes the formation of the apical tubulin network. In the present study, HDAC inhibitors induced acetylation of tubulin and recruited CGN at the membranes. We think that the induced CGN may bind to the acetylated tubulin at the membranes, although it is necessary to perform more investigation.

In addition, also the experiments are poorly described, in the immunofluorescence analysis shown in Figure 1A and 1B, it seems that authors co-stained the cells with p63 and LSR antibody, is that an error?

It is not error. In the immunofluorescence analysis of A253 cells, only polyclonal rabbit anti-p63 antibody and polyclonal anti-LSR antibody but not monoclonal mouse antibodies were work. Accordingly, we indicated one color (Alexa488) in co-stained the cells with both p63 and LSR antibodies. As p63 is expressed in the nuclei and anugulin-1 is expressed at the membranes in A253 cells, the results are correct and useful.

Finally, despite authors have started their work with the hypothesis, already published (p63 expression in human tumors and normal tissues: a tissue microarray study on10,200 tumors) that p63 is overexpressed in salivary gland carcinoma, they couldn’t even confirm this result in their system, something that do not support at all the importance of p63 in this study.

We also know that p63 is used for diagnostic utility in the histologic differentiation of salivary gland tumors. However, in p63-positive cancers including salivary gland tumors, the detailed roles of p63 remain yet unclear. We think that it is necessary to perform more investigation.

Reviewer 2 Report

The article entitled "Inhibition of HDAC....in salivary duct carcinoma" by Nakano et al; talk about the role of p63 and HDAC inhibitors in slivery duct adenocarcinoma. Authors performed the experiments in the presence of TGF Beta receptor inhibitor EW-7197 and JNK inhibitor SP600125 to give a prospective as these two pathways are associated with p63 mediated proliferation and differentiation. Authors have used GeneChip analysis to perform differential gene expression analysis post p63KD. Tight junction and apical complex protein ZO3, CGN were found to be unregulated both in p63 KD and HDAC inhibitor treated samples signifying the role of p63 and HDACs in SDC. The manuscript gave a prospective and opens the possibilities of using HDACIs in SDC, however, there are flaws which are needed to resolve. Here I am providing my views and comments.

  1. There is lack of continuity throughout the manuscript. Authors have jumped from one topic to another abruptly which makes it hard to understand and connect. Result section are explained without any rationale and conclusions.
  2. The results from GeneChip experiments need to be validated by qPCR or western blowing as I can see conflicting results in figure 3A. I don't see induction of corresponding proteins in WB after p63 KD. 
  3. Legend section of figure 1 has issues.
  4. Figure 2A suggests that p63 works downstream of HDAC as there is downregulation post HDAC inhibition, it is intriguing that the phenotype is very strong (up regulation of TJ proteins) in HDACi treated samples than p63 KD, however, the initial observation is based on p63 KD, why is this discrepancy? I would also suggest to add EMT marker proteins like ZEB1, SNAIL to get a clear picture of EMT.
  5. Result section 3.7 and figure 4F-G are poorly explained. The quality of figure need to improve.
  6. Result section 3.8 was placed without any rationale, need to explain properly.
  7. Discussion section page 16 paragraph 6, line 6, it was said that HDAc inhibition did not affect the OCR, I don't see the experiment.
  8. As usual discussion lacks connectivity.

Author Response

Response to Comments and Suggestions of Reviewer 2

The article entitled "Inhibition of HDAC....in salivary duct carcinoma" by Nakano et al; talk about the role of p63 and HDAC inhibitors in slivery duct adenocarcinoma. Authors performed the experiments in the presence of TGF Beta receptor inhibitor EW-7197 and JNK inhibitor SP600125 to give a prospective as these two pathways are associated with p63 mediated proliferation and differentiation. Authors have used GeneChip analysis to perform differential gene expression analysis post p63KD. Tight junction and apical complex protein ZO3, CGN were found to be unregulated both in p63 KD and HDAC inhibitor treated samples signifying the role of p63 and HDACs in SDC. The manuscript gave a prospective and opens the possibilities of using HDACIs in SDC, however, there are flaws which are needed to resolve. Here I am providing my views and comments.

Thank you for your interesting on our paper.

  1. There is lack of continuity throughout the manuscript. Authors have jumped from one topic to another abruptly which makes it hard to understand and connect. Result section are explained without any rationale and conclusions.

We added some sentences in all result sections.

  1. The results from GeneChip experiments need to be validated by qPCR or western blowing as I can see conflicting results in figure 3A. I don't see induction of corresponding proteins in WB after p63 KD. 

We also expected the induction of the corresponding proteins OLFL3, CY1A1, AQP7, TLR1, TLR5 and ELF3 in WB after p63 KD. We performed the WB. OLFL3 and CY1A1 were detected in WB indicated as Figure 3A, but no change was observed after p63 KD. AQP7, TLR1, TLR5 and ELF3 were not detected in WB using A-253 cells.

Although the detailed mechanisms are unclear, we think that in the A-253 cells, the changes of the proteins are weak, because OLFL3 and CY1A1 proteins are too much and TLR1, TLR5 and ELF3 are too less.

  1. Legend section of figure 1 has issues.

We rewrote it.

  1. Figure 2A suggests that p63 works downstream of HDAC as there is downregulation post HDAC inhibition, it is intriguing that the phenotype is very strong (up regulation of TJ proteins) in HDACi treated samples than p63 KD, however, the initial observation is based on p63 KD, why is this discrepancy? I would also suggest to add EMT marker proteins like ZEB1, SNAIL to get a clear picture of EMT.

We agreed with the comments. We performed the WB for an EMT marker protein SNAIL. However, the clear change of SNAIL protein was not observed. It is thought that there is not only downregulation of p63 but also the other mechanisms in upregulation of TJ proteins by treatment with HDACi. In near future, we will perform the other intercellular signaling including NF-κB. Because we found that NF-κB inhibitor curcumin could induce TJ proteins in A-253 cells (data not shown).

  1. Result section 3.7 and figure 4F-G are poorly explained. The quality of figure need to improve.

We added the schema of OCR as figure 4G and the final result sentence.

  1. Result section 3.8 was placed without any rationale, need to explain properly.

We added some rationale sentences in result section 3.8.

  1. Discussion section page 16 paragraph 6, line 6, it was said that HDAC inhibition did not affect the OCR, I don't see the experiment.

We mistook it. We added the data as supplemental Figure 2.

  1. As usual discussion lacks connectivity.

We added the connected words in discussion.

Reviewer 3 Report

I would like to thank the authors for their nice job. The study is well performed. The manuscript is clear and well written.  The figures are clear.

This manuscript adds new insights on the role of inhibition of HDAC and signal transduction in treating salivary malignancies 

Author Response

Response to Comments and Suggestions of Reviewer 3

I would like to thank the authors for their nice job. The study is well performed. The manuscript is clear and well written.  The figures are clear. This manuscript adds new insights on the role of inhibition of HDAC and signal transduction in treating salivary malignancies 

Thank you for your interesting on our paper.

Round 2

Reviewer 1 Report

Authors have performed new experiments and improved the description of their results. Although, I still think that in the discussion they should claim that the experiments performed knocking-down p63 were performed in the presence of the two inhibitors EW-7197 and SP600125. In particular, they should add this information in this sentence: "In this study, a DNA array showed that knockdown of p63 downregulated p63 mRNA and upregulated mRNAs of the TJ proteins CGN and ZO-3 and the epithelial dif- ferentiation markers OLFM3, CY1A1, AQP7, TLR1, TLR5 and ELF3. The knockdown of p63 recruited the TJ proteins CGN, ZO-3, angulin-1/LSR and OCLN at the membranes and prevented cell proliferation. The cell differentiation of salivary gland duct carcinoma cells may be accompanied by a decrease in p63 and by a shift to adhesion patterns involving tight junctions."

Moreover, authors have added new sentences at the beginning of each result section as an introduction of their results, but they did not put any reference. I think that they should add them.

Author Response

Authors have performed new experiments and improved the description of their results. 
Although, I still think that in the discussion they should claim that the experiments performed knocking-down p63 were performed in the presence of the two inhibitors EW-7197 and SP600125. In particular, they should add this information in this sentence: "In this study, a DNA array showed that knockdown of p63 downregulated p63 mRNA and upregulated mRNAs of the TJ proteins CGN and ZO-3 and the epithelial differentiation markers OLFM3, CY1A1, AQP7, TLR1, TLR5 and ELF3. The knockdown of p63 recruited the TJ proteins CGN, ZO-3, angulin1/LSR and OCLN at the membranes and prevented cell proliferation. The cell differentiation of  salivary gland duct carcinoma cells may be accompanied by a decrease in p63 and by a shift to adhesion patterns involving tight junctions."

We agree with the suggestions of the reviewer and added the words of two inhibitors EW-7197 and SP600125 in the sentences of Discussion.

Moreover, authors have added new sentences at the beginning of each result section as an introduction of their results, but they did not put any reference. I think that they should add them.

We added the references in new sentences.

Furthermore, it is known that p63 is regulated through a NF-κB signal pathway. We added the data of NF-κB inhibitor curcumin as Figure 4 and rewrote Results and Discussion.

‘’p63 may be regulated by NF-κB [5]. To investigate the effects of p63 via a NF-κB signal pathway on TJ proteins in SDC, A253 cells were treated with the NF-κB inhibitor curcumin at 10 μM under pretreatment with TGF-β type I receptor inhibitor EW-7197 and JNK inhibitor SP600125 at 10 μM. In immunocytochemical analysis, p63 disappeared from the nuclei, and angulin-1/LSR, OCLN, TRIC and CLDN-4 were recruited at the membranes by treatment with curcumin under pretreatment with or without EW-7197 and SP600125 (Figure 4A). In Western blot analysis, downregulation of p63, phosphorylated NF-κB and phosphorylated MAPK and upregulation of CLDN-4 were observed by treatment with curcumin under pretreatment with or without EW-7197 and SP600125 (Figure 4B).’

Please check the attached pdf.

Reviewer 2 Report

I could see the improvement after the revision, however, I don't see any improvement in terms of result. The HDACi treatment results are promising, while the GeneChIP results are confusing as it is not validated by alternate methods like qPCR/ Western blot. I would like to recommend authors to rewrite the paper around HDACi and perform more experiments related to NFkB and EMT. 

Author Response

1) Although we did not perform GeneChip/qPCR in the cells treated with HDACi under pretreatment with EW-7197 and SP600125, it is possible that treatment with HDACi may induce many genes. Because HDAC-targeting genes are too much in cancer cells and HDACi has multi-functions. Accordingly, we think that in the present study, it is difficult to find the targeting genes for TJ proteins and differentiation from data of GeneChip/qPCR in HDACi-treated cells.

2) As following your suggestions, we added the data of NF-κB inhibitor curcumin as Figure 4 and rewrote Results and Discussion.
‘’p63 may be regulated by NF-κB [5]. To investigate the effects of p63 via a NF-κB signal pathway on TJ proteins in SDC, A253 cells were treated with the NF-κB inhibitor curcumin at 10 μM under pretreatment with TGF-β type I receptor inhibitor EW-7197 and JNK inhibitor SP600125 at 10 μM. In immunocytochemical analysis, p63 disappeared from the nuclei, and 
angulin-1/LSR, OCLN, TRIC and CLDN-4 were recruited at the membranes by treatment with curcumin under pretreatment with or without EW-7197 and SP600125 (Figure 4A). In Western blot analysis, downregulation of p63, phosphorylated NF-κB and phosphorylated MAPK and upregulation of CLDN-4 were observed by treatment with curcumin under pretreatment with or without EW-7197 and SP600125 (Figure 4B).’

3) In the present study, we did not perform EMT experiments because EMT was not mainly discussion. We promise to perform it in near future and will report the data in the next paper

Please check the attached pdf. 

Round 3

Reviewer 2 Report

I am satisfied with the explanation and curcumin experiment.